# Improved Mathematical Approach for Modeling Sport Differential Mechanism

**Maksym Diachuk and Said M. Easa \***

Department of Civil Engineering, Ryerson University, 350 Victoria Street, Toronto, ON M5B2K3, Canada;
maksym.diachuk@ryerson.ca
* Correspondence: seasa@ryerson.ca

**Abstract:** Improved mathematical and simulation modes of the active differential mechanism (DM) with controllable torque redistribution would better contribute to developing intelligent vehicle transmissions. The issue is caused by actualizing the precise steerability control using advanced automated transmissions, allowing torque vectoring for all-wheel-drive vehicles and ensuring an option for correcting the vehicle trajectory. This paper presents an alternative mathematical method for obtaining differential equations for modeling vehicle transmission components and its implementation for simulating the Audi sport DM. First, the steerability issues of sport DM technology are discussed, and the sport DM design is described in detail. Then, a mathematical approach is proposed that includes three types of equation systems: generalized dynamics equations, kinematic constraint equations, and gearing condition equations. The approach also considers the flexibility of the clutch's frictional pack, friction torque, lockup condition, and piston dynamics. Finally, a Simulink model that reflects the DM operation and calculation procedures is developed. A series of simulations of the sport DM operation with forcible torque distribution is carried out. The results show that the proposed mathematical model is universal, efficient, and accurate.

**Keywords:** sport differential; torque vectoring; friction clutch; vehicle kinematics

## 1. Introduction

The integrated vehicle control affords maximum vehicle performance, handling accuracy, and safety. Notably, the distributed torque technology for all-wheel-drive (AWD) vehicles provides the optimal mode of operation for each wheel individually. The redistribution of traction forces reduces the lateral slip process due to the partial compensation of lateral speeds, which increases control accuracy and motion stability. In addition, the distributed torque technology adjusts the exactness of the vehicle trajectory, which is especially essential for autonomous vehicles. Today, many vehicle manufacturers use the sport differential technology that does not require activating the inner wheels' brakes during the curvilinear motion, unlike the pure torque vectoring (TV). Electric and hydraulic drives often carry out the control of such differentials.

The characteristics of selected studies on vehicle differential mechanism (DM) are presented in Table 1. As noted, the research field of DM modeling is quite broad, spanning different methods, approaches, and study areas. Nevertheless, there is a lack of research on modeling active differentials with the electro-hydraulic drive since most papers have focused on active limited-slip differential (ALSD). Additionally, there is a lack of comprehensive studies that generalize the active differential control algorithms to solve the problems of motion stability, understeer compensation, and the increase in vehicle passability [1]. The math models and equation systems are primarily classic and do not imply systematization into one method. Some math models are too complicated to be used in vehicle dynamics and focused on pure mechanical objectives.

**Table 1.** Characteristics of selected studies on vehicle differential mechanisms.

| Reference | Topic | Features |
|:---:|:---:|:---|
| [1] | Limited slip, self-locking TORSEN DM | Actualize issues of differentials and their influence on vehicle dynamics. Formed cornering moment in conditions of different slips. Compared DM to the difference in angular speeds loading torques. |
| [2] | Convectional, LSD, controlled LSD | Introduced torsional stiffness and lash of mechanical gearing. Matrix approach and simulation schemes. Developed detailed and simplified DM models. Combined DM and vehicle modeling for testing the cornering effect. |
| [3] | Active DM, Torque Vectoring | Developed a unified math model for active differential dynamics. Various DM designs and levels of model complexity are used. Restricted applicability as estimated time response is needed. |
| [4] | Active LSD (ALSD) | Investigated driveline and tire model effects on the ALSD performance. ALSD design includes friction clutches for transmitting the torque. Energy losses math models and Simulink tools are included. |
| [5] | Active LSD | Developed a control algorithm for a rear-wheel-drive sport vehicle. Compared ALSD impact on vehicle model behavior. Assessed ALSD influence on driver workload. |
| [6] | Active LSD | TV differential mechanism with electrohydraulic actuation. Race car model with 7 DOFs and low ground effect. Implemented nested control loop for the actuation system. |
| [7] | TV differential, electronic stability control | Nonlinear vehicle model based on CarSim software. TV Differential with two series of planetary gears. Electronic stability model with three-layer Integrated control system. Unscented Kalman filter and controller based on BA optimization. |
| [8] | TORSEN DM | Three-dimensional cylindrical joint model with clearance, misalignment, and friction. Matrix dynamics system including holonomic bi-lateral constraints. |
| [9] | Inter-wheel differential | DM with power balance and kinematic relations among three shafts. Three differential equations; no efficiencies or changes in power flows. |
| [10] | TV differential | New TV differential based on a *Ravigneaux* gear set. Two different speed ratios with only one pair of gear sets. *SimulationX* software is used to test maneuverability and steerability. |
| [11] | Original DM design | DM for TV concept; design combines inner gears. Math model includes dynamics and kinematic constraints equations. Losses, efficiencies, and power flow direction are neglected. |
| [12] | Asymmetric differential | Developed two DM variants (conic and planetary gear). Dynamics and constraints; static friction and limited-slip functionality. Overcomes simulation problems of discontinuity at zero angular speed. |
| [13] | TORSEN DM | Multibody simulation using nonlinear FEM. Flexible gear pair joints and contact conditions are used as constraints. AWD model for estimating torque redistribution. |
| [14] | Controllable DM | DM based on the magnetorheological fluid to realize the locking state. Torque, power balance, and kinematic constraint equations. Double-controller scheme including extended Kalman filter. SIL and HIL modeling using experimental prototypes. |

This paper presents a universal mathematical approach for modeling vehicle transmission components and its implementation for simulating the Audi sport DM, especially for hardware in the loop (HIL) and software in the loop (SIL) modeling. The specific contributions of the paper are: (1) to describe the steerability issues and the design of sport DM technology, (2) to propose an alternative mathematical method for obtaining differential equations that describe the dynamics of rotational mechanical systems, includ-

ing generalized dynamics equations, kinematic constraint equations, gearing condition equations, and frictional clutch, and (3) to compose a Simulink model that reflects the DM operation and calculation procedures.

## 2. Background on Sport DM Technology

### 2.1. Steerability Issues

Consider the case of a passenger vehicle's curvilinear motion (Figure 1a). It is almost impossible to ensure the ideal instant turn with the neutral steer in actual conditions due to both tires' lateral elasticity and inevitable slip in the contact spots. As known well in this regard, two distinctive phenomena may occur-understeer and oversteer. Both processes are characterized by an intense lateral component of the instantaneous velocity in the tire-road contact spot caused by the sideslip. These phenomena are associated with the distribution of vertical reactions along the vehicle axles. From a physical point of view, it is desirable to have approximately the same slip conditions for all tires to provide predictable control. Thus, AWD vehicles can adjust the traction forces to compensate for the slip individually. If a vehicle is designed so that the front axle bears a larger mass, then it has the understeer tendency. In this case, as shown in Figure 1a, the instantaneous rotation center $O$ is located behind the rear axle [15]. Thus, the transversal components of the instantaneous velocities appear in all wheels' contact patches. The curvature of the motion trajectory decreases compared to the required one to ensure trajectory stability. As a result, to compensate for the lacking trajectory curvature, it is necessary to permanently increase the steered wheels' angles or reduce the cruise speed.

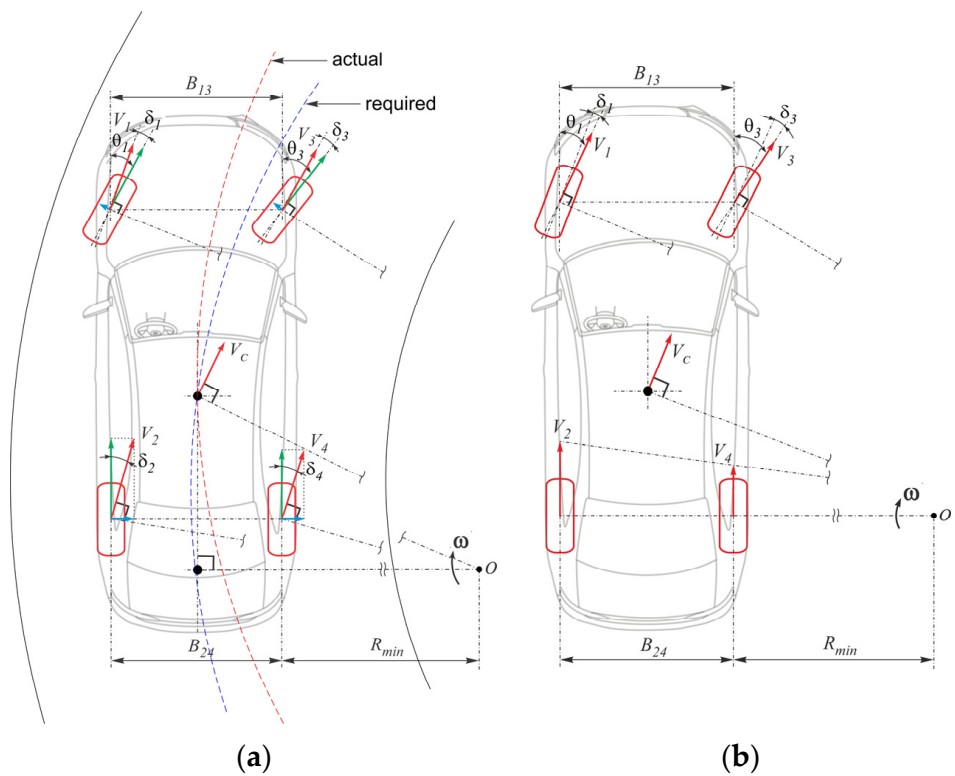

**(a)** **(b)**

**Figure 1.** Steerability cases: (**a**) understeer and (**b**) neutral steer.

It is possible to create an additional yaw moment in the traction mode by changing longitudinal reactions on the vehicle's wheels. However, if symmetric (open) differentials are used in transmissions, the responses on drive semi-axles are practically set equal, which does not affect the yaw moment. Thus, the prerequisites emerge for controlling the movement accuracy or tracking a planned trajectory by redistributing the torque between wheels, which may be achieved, among other things, by active differentials. Since the

difference between traction forces on the same axle wheels, an additional yaw moment occurs, decreasing the slip angles and approaching an instant turn to a scheme close to the neutral steer (Figure 1b). At the same time, the tires' lateral forces can reach larger values and ensure control accuracy and trajectory stability (strict tracking) with a smaller steering angle.

## 2.2. Design of Audi DM

Several limited-slip differentials distribute the torques depending on the wheel operational mode. However, their redistribution concept implies that the greater torque is passed to a wheel that is either lagging or has better adhesive conditions. As a result, the sport differential technology must transmit a more significant torque value to an outrunning axle, which causes additional cornering (yaw moment) relative to the vertical axis passing through the mass center. Such a solution can be obtained using other planetary gears (BMW, Honda) or, for example, two-step internal gearing (Audi) controlled by friction clutches with electric or hydraulic drives. Consider the design scheme and functioning of Audi Sport Differential [16] shown in Figure 2. The primary open information may be taken from Audi Sport Differential Technology[17].

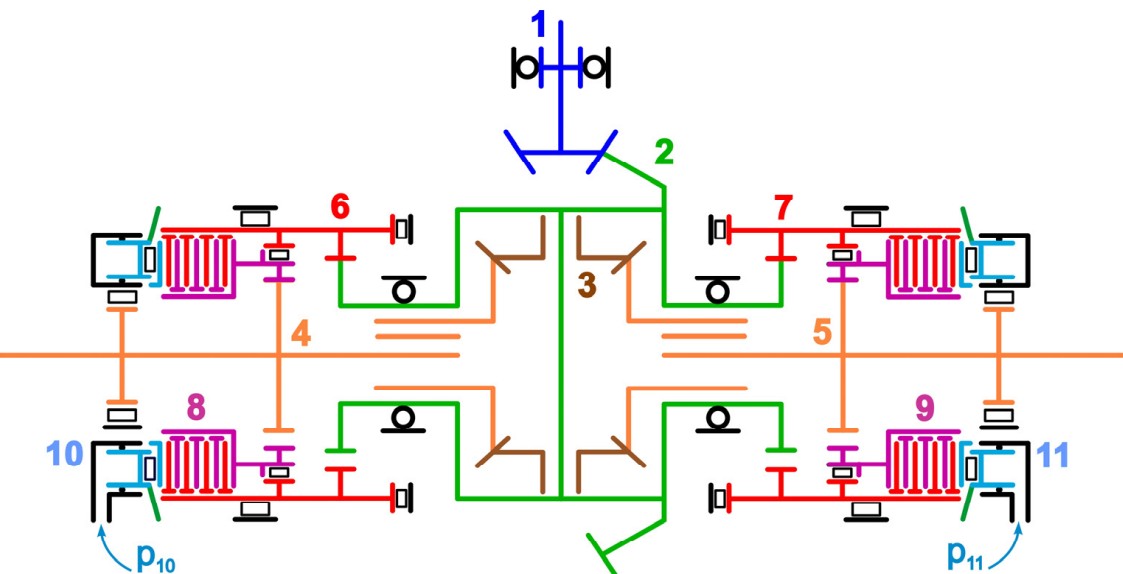

**Figure 2.** Scheme of Audi Sport Differential.

The drive is carried out over the final gear pinion 1. The ring gear is rigidly connected to the differential carrier 2. Satellites 3, rotating around the axis fixed in the differential carrier, interact with side gears that drive the output axles 4 and 5 by the slots. The differential corps (carrier) has gear rims on the end sides for driving by internal gearing the coupling halves 6 and 7, in which rotational axes are respectively shifted relative to the carrier rotational axis. Using the frictional packs, the half-clutches 6 and 7 can drive the half-clutches 8 and 9, respectively, which are connected by an internal gearing to the output shafts 4 and 5. Toroidal hydraulic cylinders 10 and 11 are installed from each side to act on the clutch packs using the pressure $p_{10}$ and $p_{11}$. Thus, by activating the required hydraulic cylinder, part of the carrier torque may be passed to the needed semi-axle using the frictional adhesion between the half-couplings over the two-step internal gearing.

Based on the sequence of links transmitting torque, the arrays of permanent liaisons $\mathbf{L}$ (Figure 3), friction couples $\mathbf{C}$, and vectors $\mathbf{i}$ of ratios and $\boldsymbol{\eta}_G$ of gearing efficiencies can be introduced. Moreover, if the numbers of each detail couple are denoted as a column-vector $\mathbf{L}_k$ (where $k = 1, \ldots, m$, and $m$ is the number of pairs), the sequential disposition of the conjugate details may be rearranged in a row vector $\mathbf{l}$, as follows:

$$L = \begin{pmatrix} 1 & 2 & 2 & 2 & 3 & 3 & 4 & 5 \\ 2 & 3 & 6 & 7 & 4 & 5 & 8 & 9 \end{pmatrix}, \ l = \begin{pmatrix} L_1^T & \cdots & L_m^T \end{pmatrix}, \ C = \begin{pmatrix} 6 & 7 \\ 8 & 9 \end{pmatrix} \tag{1}$$

$$\begin{aligned} i &= \begin{pmatrix} i_{12} & i_{23} & i_{26} & i_{27} & i_{34} & i_{35} & i_{48} & i_{59} \end{pmatrix}^T, \\ \eta_G &= \begin{pmatrix} \eta_{12} & \eta_{23} & \eta_{26} & \eta_{27} & \eta_{34} & \eta_{35} & \eta_{48} & \eta_{59} \end{pmatrix}^T \end{aligned} \tag{2}$$

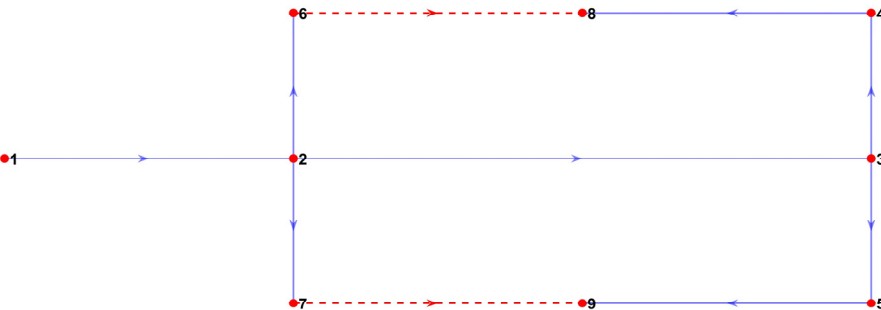

**Figure 3.** Graph of mechanical links.

The friction clutch's slip degree affects the amount of additional torque withdrawn from the differential carrier and the fact that, as shown in Figure 1, for the general case of curvilinear motion, the angular speeds of the coupling halves 6, 7 and 8, 9 should be different. Thus, the pressure in the hydraulic cylinders must be adjusted in order, on the one hand, to maintain the ratio of wheels' angular speeds required during curvilinear motion and, on the other hand, to prevent the clutch lock-up. Requirements for passing the greater torque and high revolutions to the external wheel impose specific gearing ratios on the clutches 6–8 and 7–9. Consider this situation using the example of parameters in Figure 1b, which corresponds to ideal cornering with minimal sideslip. Assuming that a clutch state close to complete locking is possible only for the variant of turning with a minimal radius, it is possible to determine the gear ratios for the drive of clutches' half-couplings. Determine the difference in angular speeds of the rear axle wheels. Their linear speeds are given by

$$V_4 = \omega R_{min}, \ V_2 = \omega (R_{min} + B_{24}) \tag{3}$$

where $\omega$ = instantaneous angular rate of the turn, and $B_{24}$ = transversal base of rear wheels.

Then, the rear wheels' angular speeds $\omega_{w4}$, $\omega_{w2}$ can be tied with the parameters of the turn kinematics as:

$$\omega_{w4} = \frac{V_4}{r_{e4}} = \frac{\omega R_{min}}{r_{e4}}, \ \omega_{w2} = \frac{V_2}{r_{e2}} = \frac{\omega (R_{min} + B_{24})}{r_{e2}} \tag{4}$$

where $r_{e4}$, $r_{e2}$ = wheels' effective radii (almost equal in most cases).

The angular velocities' ratio, considering the designations in Figs. 1a and 1b, is estimated as

$$k_\omega = \frac{\omega_{w2}}{\omega_{w4}} = \frac{\omega_4}{\omega_5} = \frac{\omega (R_{min} + B_{24})}{r_{w2}} \frac{r_{w4}}{\omega R_{min}} \approx \frac{R_{min} + B_{24}}{R_{min}} \tag{5}$$

where $\omega_{w2} = \omega_4$ and $\omega_{w4} = \omega_5$ are the angular velocities of the rear wheels (Figure 1b) and the corresponding axles (Figure 2).

For the case of the angular speed distribution based on the symmetric differential's kinematics (components 2 and 3 in Figure 2), the following condition must be satisfied:

$$2\omega_2 = \omega_{w2} + \omega_{w4} = \omega_{w2} + \omega_{w2}/k_\omega = (1 + 1/k_\omega)\omega_{w2} \tag{6}$$

but also

$$2\omega_2 = \omega_4 + \omega_5 = \omega_4 + \omega_4/k_\omega = (1 + 1/k_\omega)\omega_4$$

Then, the required angular speed of the differential carrier is given by

$$\omega_2 = \frac{(1+1/k_\omega)}{2}\omega_4 = \frac{(1+1/k_\omega)}{2}\omega_{w2} \tag{7}$$

This determines the needed ratio between the steps of the half-couplings when a friction clutch locks them. That is,

$$i_{24} = \frac{\omega_2}{\omega_4} = i_{26}i_{84} = \frac{\omega_2}{\omega_6}\frac{\omega_8}{\omega_4} = \frac{(1+1/k_\omega)}{2} \tag{8}$$

Since the torque is transmitted over the friction clutch in two steps of the internal gearing, then

$$i_{26} = \omega_2/\omega_6 = z_6/z_2, \ i_{84} = \omega_8/\omega_4 = z_4/z_8 = 1/i_{48} \tag{9}$$

Taking the integer teeth numbers for the gear rims of the half-couplings, the final values are obtained as shown in Table 2.

**Table 2.** Data for determining the ratios for gears of half-couplings.

| $B_{24}$ | $R_{min}$ | $i_{24}$ | $z_2$ | $z_6$ | $i_{26}$ | $z_4$ | $z_8$ | $i_{84}$ |
|---|---|---|---|---|---|---|---|---|
| 1.551 | 5.8 | 0.887 | 33 | 41 | 1.242 | 25 | 35 | 0.714 |

The obtained value of $i_{24}$ shows that the difference between the angular speeds of the differential's carrier and the outer rear wheel differs only by about 11%, even at the maximum steering angle. Thus, the angular speeds of the half-couplings can be compared when moving with a lesser curvature, as follows

$$\omega_6 = \frac{\omega_2}{i_{26}} = \frac{\omega_2}{1.242} = 0.805\omega_2, \ \omega_8 = \omega_4 i_{84} = 0.714\omega_4 \tag{10}$$

If, for instance, the movement is close to a straight line, then $\omega_2 = \omega_4$, and it follows from Equation (10) that $\omega_6 > \omega_8$ by about 11%. For all cases when the instantaneous curvature radius is greater than $R_{min}$, it remains true that $\omega_6 > \omega_8$, which corresponds to the need for a friction clutch's slip that regulates the required instantaneous radius of the vehicle's turn. In this case, half-couplings 6 or 7 will be driving, depending on the activation order. It is also evident that at turning with the maximum angles of steered wheels, $\omega_6 \approx \omega_8$ and consequently, the friction clutch can be locked. As it follows, if the pressure in the hydraulic cylinder leads to locking the clutch, the ratio of the rear wheels' angular speeds will correspond to the kinematics of turning with a minimum radius even for straight motion.

### 3. Integrated Mathematical Model

This section presents the approach for generalizing the rotational dynamics equations followed by the integration of the whole mathematical model. The research methodology is shown in Figure 4 and explains the steps needed to build and test the model (the respective subsection of each step is shown in parentheses). First, three types of equations are considered separately: generalized dynamics equations, kinematic constraint equations, and gearing condition equations. Then, appending the equations for translationally moving elements, the math model of the entire mechanism is composed, considering friction clutch (friction torque, lockup condition, and piston dynamics). Finally, a series of virtual tests were conducted to validate the sport differential's unique properties and the simulation approach in general.

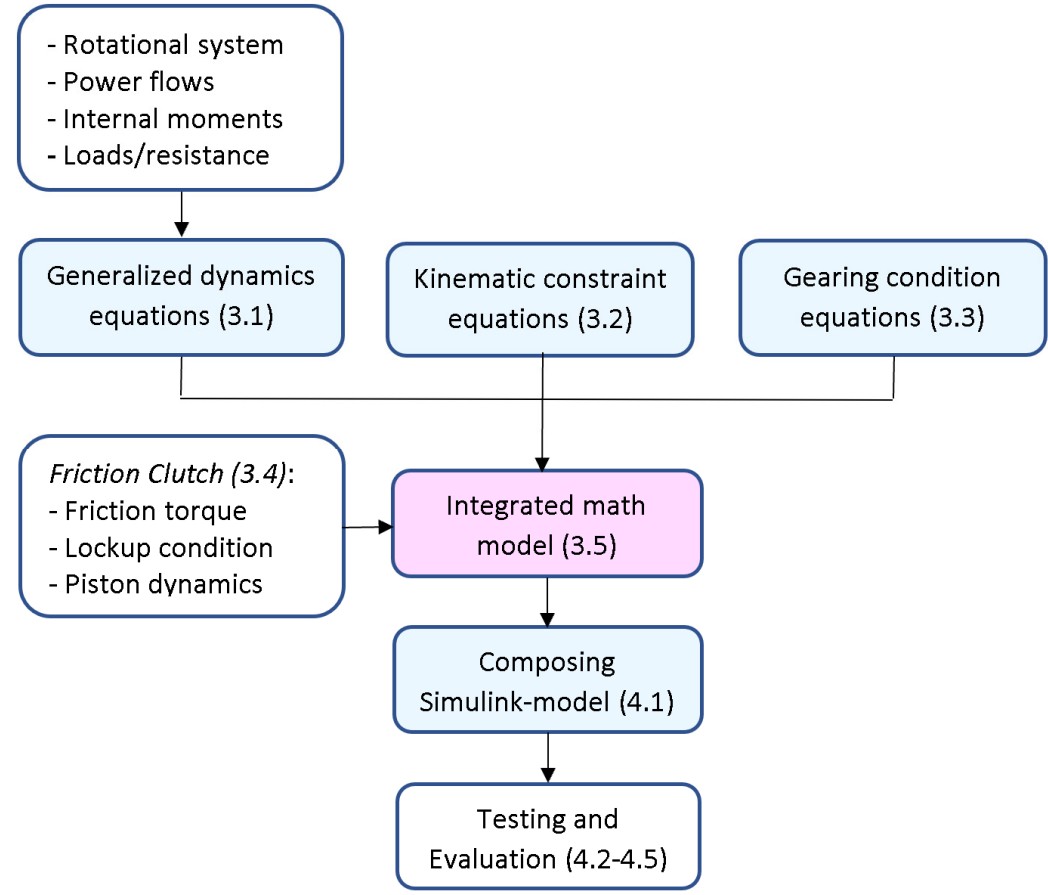

**Figure 4.** Process of integrated mathematical model.

Each structural component of the Audi sport differential can be described by a system of equations, including dynamics balance, kinematic constraints, and contact force relations. Figure 5 depicts all the design elements accompanied with the needed geometric parameters and the force factors acting between the details.

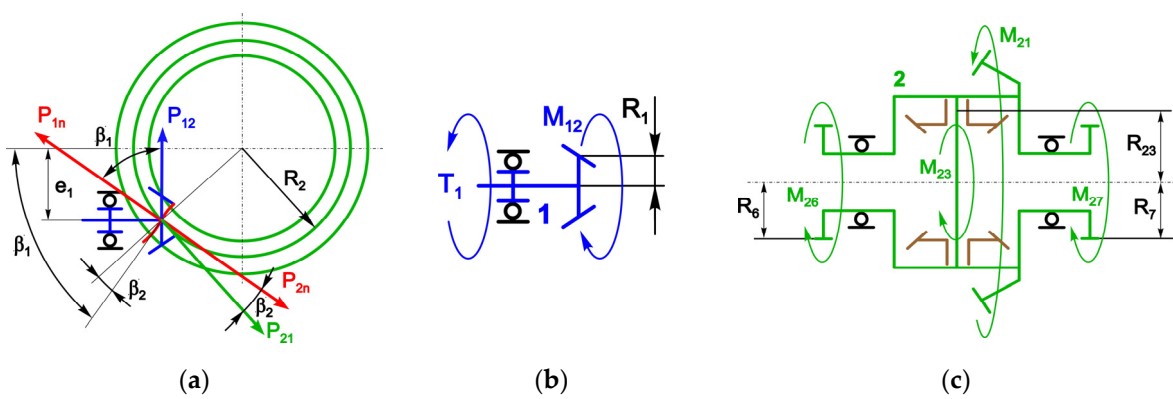

(**a**)                                     (**b**)                                     (**c**)

**Figure 5.** *Cont.*

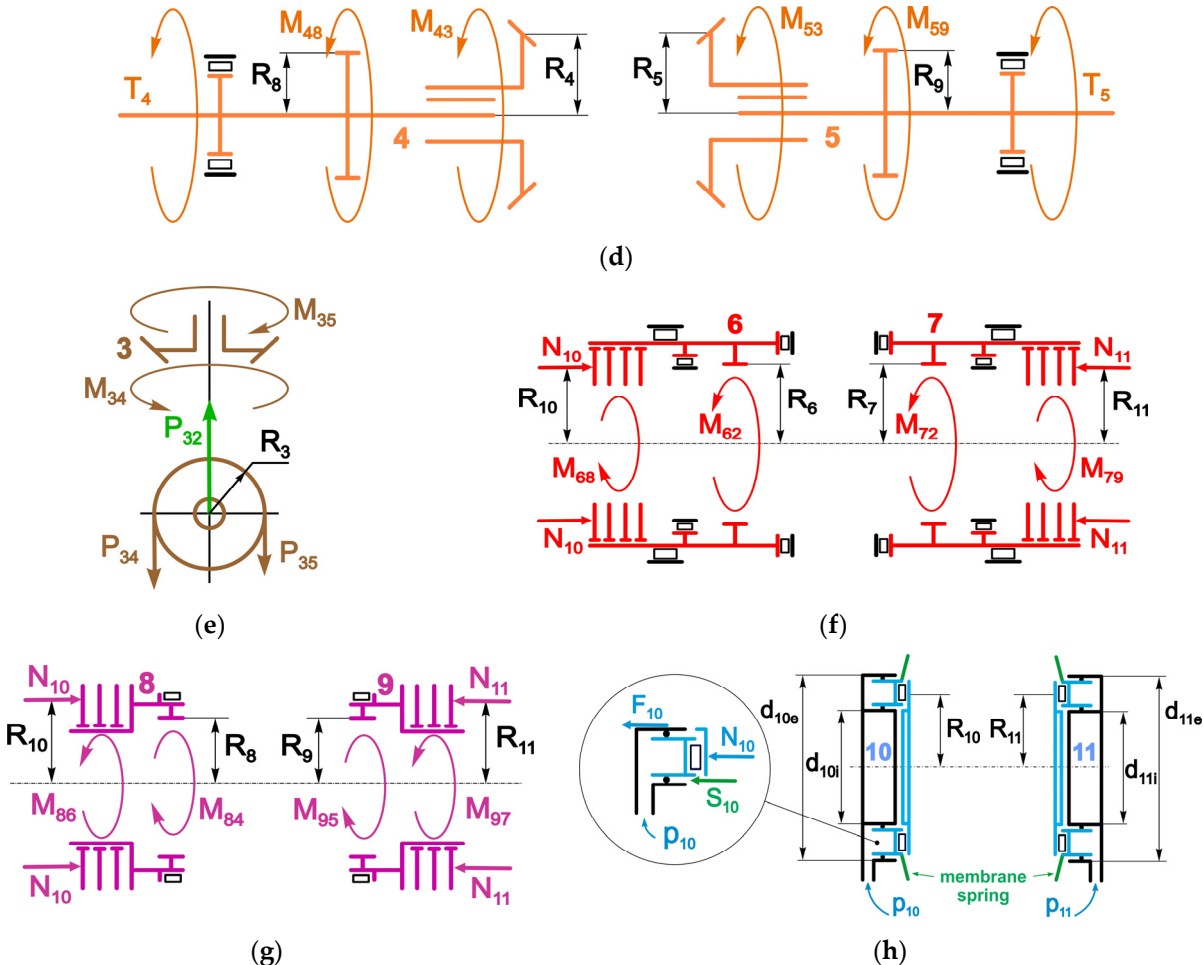

**Figure 5.** Parameters and force factors acting on the parts of the sport differential: (**a**) final drive's hypoid gearing, (**b**) final drive pinion, (**c**) carrier body, (**d**) satellite, (**e**) side gears and output shafts, (**f**) clutches' leading half-couplings, (**g**) clutches' driven half-couplings, and (**h**) clutches' hydraulic cylinders.

The rear-axle final drive as the component of Audi Sport Differential is represented by the hypoid gearing, unlike, for instance, the pure conic final drive of the vehicle front axle, and shown in Figure 5a. Among the advantages of such a design solution, many aspects may be listed, such as a higher gear ratio with smaller gear ring size, increased teeth strength, and reduced noise. The main drawbacks include working with teeth sliding that reduces the gearing efficiency and requires special oils for high-pressure mechanical contacts. According to the scheme in Figure 5a, the force of contact reaction may be decomposed as [18].

$$P_{21} = P_n \cos(\beta_1), \ P_{12} = P_n \cos(\beta_2) \tag{11}$$

where the angles $\beta_1$ and $\beta_2$ are conditioned by the eccentricity $e_1$ and teeth curvature. It is recommended that $\beta_1 = 45°$ to $50°$ and $\beta_2 = 45°$ to $50°$. Thus, the hypoid final drive ratio can be expressed as

$$i_{12} = \frac{\omega_1}{\omega_2} = \frac{M_{21}}{M_{12}} = \frac{D_\omega P_{12}}{d_\omega P_{21}} = \frac{D_\omega P_n \cos(\beta_2)}{d_\omega P_n \cos(\beta_1)} = \frac{D_\omega}{d_\omega} k_{12}, \ k_{12} = \frac{\cos(\beta_2)}{\cos(\beta_1)} = 1.2 \ldots 1.5 \tag{12}$$

where $d_\omega$ and $D_\omega$ = reference (pitch) diameters of the pinion and ring, respectively.

The hypoid gearing efficiency and teeth sliding speed may be calculated as

$$\eta_{12} = \frac{1 + \mu_{12} \tan(\beta_2)}{1 + \mu_{12} \tan(\beta_1)}, v_s = v_1 \frac{\sin(\beta_1 - \beta_2)}{\sin(\beta_2)} \tag{13}$$

where $\mu_{12}$ = coefficient of teeth friction ($\mu_{12} = 0.05 \ldots 0.1$, $\eta_{12} = 0.96 \ldots 0.97$), and $v_1$ = pinion peripheral speed.

Consequently, the torque relation for the final drive gearing may be written for the two cases, depending on the power flow passing through pinion and gear ring, as

$$M_{21} + M_{12} i_{12} \eta_{12} = 0, \; M_{21} \eta_{21} + M_{12} i_{12} = 0 \tag{14}$$

*3.1. Generalization of Dynamics Equations*

3.1.1. Rotational System

Using Figure 5, the system of differential equations was combined for each design element[3], considering generalized states of power flows between the parts. Assuming that the internal moments are unknown, the system may be represented in the form of the extended left side as

$$
\begin{cases}
I_1 \varepsilon_1 \eta_{1B} - M_{12} \eta_{1B}^{(c)} = T_1 \eta_{1B}^{(s)} + V_1 \\
I_2 \varepsilon_2 \eta_{2B} - M_{21} \eta_{2B}^{(s)} - M_{23} \eta_{2B}^{(c)} - M_{26} - M_{27} = V_2 \\
n_s I_3 \varepsilon_3 \eta_{3B} - n_s M_{34} \eta_{3B} - n_s M_{35} \eta_{3B} = n_s V_3 \\
I_4 \varepsilon_4 \eta_{4B} - M_{43} \eta_{4B}^{(s)} - M_{48} = T_4 \eta_{4B}^{(c)} + V_4 \\
I_5 \varepsilon_5 \eta_{5B} - M_{53} \eta_{5B}^{(s)} - M_{59} = T_5 \eta_{5B}^{(c)} + V_5 \\
I_6 \varepsilon_6 \eta_{6B} - M_{62} \eta_{6B}^{(s)} = M_{68} \eta_{6B}^{(c)} + V_6 \\
I_7 \varepsilon_7 \eta_{7B} - M_{72} \eta_{7B}^{(s)} = M_{79} \eta_{7B}^{(c)} + V_7 \\
I_8 \varepsilon_8 \eta_{8B} - M_{84} \eta_{8B}^{(s)} = M_{86} \eta_{8B}^{(c)} + V_8 \\
I_9 \varepsilon_9 \eta_{9B} - M_{95} \eta_{9B}^{(s)} = M_{97} \eta_{9B}^{(c)} + V_9
\end{cases}
\tag{15}
$$

where $I_n$ = moment of inertia, $\varepsilon_n$ = angular acceleration, $\eta_{nB}$ = bearing efficiency, $M_{nk}$ = internal and external moments ($k$ is the position of counteracting element), $V_n$ = moment of viscous losses, $T_n$ = external torque, $n_s$ = number of satellites, $n$ = position of element ($n = \{1, \ldots, 9\}$), and $s$ and $c$—upper indexes for meaning state and converse state of transmitting the power, respectively.

Let $\boldsymbol{I}$ be the vector of inertias, $\boldsymbol{\eta}_B$ be the vector of pure bearing efficiencies, $\boldsymbol{\eta}_T$ be the vector of external torques' efficiencies, $\boldsymbol{\eta}_M$ be the vector of friction torques' efficiencies, and $\boldsymbol{n}_s$ be the vector of satellite quantity. Then,

$$
\boldsymbol{I} = \begin{pmatrix} I_1 \\ I_2 \\ I_3 \\ I_4 \\ I_5 \\ I_6 \\ I_7 \\ I_8 \\ I_9 \end{pmatrix}, \boldsymbol{\eta}_B = \begin{pmatrix} \eta_{1B} \\ \eta_{2B} \\ \eta_{3B} \\ \eta_{4B} \\ \eta_{5B} \\ \eta_{6B} \\ \eta_{7B} \\ \eta_{8B} \\ \eta_{9B} \end{pmatrix}, \boldsymbol{\eta}_T = \begin{pmatrix} \eta_{1B}^{(s)} \\ 1 \\ 1 \\ \eta_{4B}^{(c)} \\ \eta_{5B}^{(c)} \\ 1 \\ 1 \\ 1 \\ 1 \end{pmatrix}, \boldsymbol{\eta}_M = \begin{pmatrix} 1 \\ 1 \\ 1 \\ 1 \\ 1 \\ \eta_{6B}^{(c)} \\ \eta_{7B}^{(c)} \\ \eta_{8B}^{(c)} \\ \eta_{9B}^{(c)} \end{pmatrix}, \boldsymbol{n}_s = \begin{pmatrix} 1 \\ 1 \\ n_s \\ 1 \\ 1 \\ 1 \\ 1 \\ 1 \\ 1 \end{pmatrix} \tag{16}
$$

Then, the matrix of inertia influence $I_D$ can be rewritten as

$$I_D = diag(I) \, diag(n_s) \, diag(\eta_B) \tag{17}$$

Note that the proposed system of Equation (15) is represented by equations based on the one universal equation of rotational dynamics but with a different set of unknown variables in each equation. This approach does not require complying with the variables' signs. This contributes to simplifying the general view of the system by using the matrix technique to automate the definition of variables and their signs.

### 3.1.2. Power Flows and Efficiencies

The states (*s*) and (*c*) in Equation (15) are mutually opposite and caused by different events for various design elements. Thus, the main linked components of the differential $k$ = {1, 2, 4, 5} can transmit the direct (*d*) and reverse (*r*) power flow relative to the order of nodes in the vector $L$, and, consequently, the possible states correspond to the following combinations

$$\eta_{kB}^{(s)} \in \left\{ \eta_{kB}^{(d)}, \ \eta_{kB}^{(r)} \right\}, \eta_{kB}^{(c)} \in \left\{ \eta_{kB}^{(r)}, \ \eta_{kB}^{(d)} \right\} \tag{18}$$

That is, if the state (*s*) is switched to (*d*), then state (*c*) is shifted to (*r*) and vice versa. In turn, the substitution of the values is carried out as follows

$$\eta_{kB}^{(s)} = \eta_{kB}^{(d)} = \eta_{kB} \Rightarrow \eta_{kB}^{(c)} = \eta_{kB}^{(r)} = 1 \, , \, \eta_{kB}^{(s)} = \eta_{kB}^{(r)} = 1 \Rightarrow \eta_{kB}^{(c)} = \eta_{kB}^{(d)} = \eta_{kB} \tag{19}$$

Based on the power flow, the following prerequisites may determine the direction. Suppose the power is transmitted through the drivetrain to the wheels. In that case, the input flow must exceed the algebraic sum of the output power flows regardless of the internal mechanism state since the mechanical connections themselves already consider the ratio of the input and output powers. Then, the direct power flow corresponds to the condition given by

$$|T_1\omega_1| > |T_4\omega_4 + T_5\omega_5| \tag{20}$$

which means the state (*s*) has been switched to the (*d*).

On the other hand, if the wheels drive the semi-axles, the power is returned to the transmission and the flow becomes reverse. Another situation is tied with redistributing the external powers between the wheels when their signs are opposite, and the modules differ slightly. In these cases, the condition is

$$|T_4\omega_4 + T_3\omega_3| \geq |T_1\omega_1| \tag{21}$$

which corresponds to the switching state (*s*) to (*r*).

There is a different picture with the drive of the friction clutches' components corresponding to the elements $k$ = {6, 7, 8, 9} in Equation (15). In this case, in each element, the direct/reverse state in each element can be changed independently of the general power flow direction. That is, states (*s*) and (*c*) can take on direct and reverse modes (*d*) and (*r*) depending on whether a flow corresponds to the natural order of nodes while transmitting power or to the reverse order (Figure 3). Thus, in this case, the math approach, in this case, is equal to Equation (19).

Note that each friction clutch can be activated individually, withdrawing some torque from the carrier and returning to it. Elements $k$ = {6, 7} are initially in the direct phase, transmitting torques from the carrier even in the case of disengaged clutches. In this case, the condition for transmitting power from the carrier can be represented as

$$|M_{62}| - |M_{86}| > 0, \ |M_{72}| - |M_{97}| > 0 \tag{22}$$

Elements $k = \{8, 9\}$ are initially in the passive phase (idle). The condition for activating the additional torque on a semi-axle can be expressed as

$$|M_{86}| - |M_{84}| > 0, \ |M_{97}| - |M_{95}| > 0 \tag{23}$$

The values of $M_{86}$ and $M_{97}$ are calculated by the dependencies for frictional moments considered next.

Denote the vectors of all the internal moments $M$, $N_s$ of the satellite quantity and bearing efficiencies depending on the states, as follows: $\eta_{BM}$ = vector for internal links, $\eta_{BS}$ = vector for satellites, $\eta_{BC}$ = vector for clutches. Vector $N_s$ can be obtained by changing values in vector $l$ Equation (1) with 1, except for positions equal to numeric 3 (satellites), which are replaced with the number of satellites $n_s$. The vector $\eta_{BS}$ structurally corresponds to the vector $N_s$ but contains efficiency $\eta_{3B}$ instead of $n_s$. Thus,

$$
M = \begin{pmatrix} M_{12} \\ M_{21} \\ M_{23} \\ M_{32} \\ M_{26} \\ M_{62} \\ M_{27} \\ M_{72} \\ M_{34} \\ M_{43} \\ M_{35} \\ M_{53} \\ M_{48} \\ M_{84} \\ M_{59} \\ M_{95} \end{pmatrix}, \ 
N_s = \begin{pmatrix} 1 \\ 1 \\ 1 \\ n_s \\ 1 \\ 1 \\ 1 \\ 1 \\ n_s \\ 1 \\ n_s \\ 1 \\ 1 \\ 1 \\ 1 \\ 1 \end{pmatrix}, \ 
\eta_{BM} = \begin{pmatrix} \eta_{1B}^{(c)} \\ \eta_{2B}^{(s)} \\ \eta_{2B}^{(c)} \\ 1 \\ 1 \\ 1 \\ 1 \\ 1 \\ 1 \\ \eta_{4B}^{(s)} \\ 1 \\ \eta_{5B}^{(s)} \\ 1 \\ 1 \\ 1 \\ 1 \end{pmatrix}, \ 
\eta_{BS} = \begin{pmatrix} 1 \\ 1 \\ 1 \\ \eta_{3B} \\ 1 \\ 1 \\ 1 \\ 1 \\ \eta_{3B} \\ 1 \\ \eta_{3B} \\ 1 \\ 1 \\ 1 \\ 1 \\ 1 \end{pmatrix}, \ 
\eta_{BC} = \begin{pmatrix} 1 \\ 1 \\ 1 \\ 1 \\ 1 \\ \eta_{6B}^{(s)} \\ 1 \\ \eta_{7B}^{(s)} \\ 1 \\ 1 \\ 1 \\ 1 \\ \eta_{8B}^{(s)} \\ 1 \\ \eta_{9B}^{(s)} \end{pmatrix} \tag{24}
$$

The matrix $H_B$ is obtained using the vectors $\eta_{BM}$, $\eta_{BS}$, and $\eta_{BC}$, and $H_T$ in terms of $\eta_T$ and $\eta_M$

$$H_B = diag(\eta_{BM})diag(\eta_{BS})diag(\eta_{BC}), \ H_T = diag(\eta_T)diag(\eta_M) \tag{25}$$

The conditions described above facilitate managing the switching between vectors in the simulation model to reflect mechanical losses according to the power relation on the DM shafts.

### 3.1.3. Internal Moments

Suppose that the vector $M$ components are initially unknown, which allows us to consider them on the system's left side of Equation (15). Each pair $(M_{kl}, M_{lk})$ reflects the force interaction of conjugated parts. This matrix is obtained by comparing the corresponding equations, and a logical matrix $E$ is introduced. To obtain it, the element-wise comparison of two matrices should be carried out. One matrix is obtained by repeating the column-vector of the parts' serial numbers $(1, \ldots, n)$ the number of times corresponding to the doubled number $(2 \cdot m)$ of the connections in the matrix $L$. The second matrix is obtained by repeating the row-vector $l$ from Equation (1) the number of times equal to the number of details $(n)$. However, pair 2–3 does not provide an absolute kinematic connection between the carrier and the satellites. The moments act in different planes, so the element corresponding to

row 3 and column 2·2 = 4 must be zeroed in matrix $E$. Thus, the left part of the system of Equation (15) associated with the unknown moments is represented in matrix form as

$$E_D = -E \, diag(N_s) \, diag(H_B) \tag{26}$$

### 3.1.4. Loads and Resistance

After introducing the vectors $v$ of viscous resistance[13] coefficients, $V$ of viscous moments, $\omega_h$ of housing angular speeds, and $\omega_r$ of relative angular speeds, then

$$v = \begin{pmatrix} v_1 \\ v_2 \\ v_3 \\ v_4 \\ v_5 \\ v_6 \\ v_7 \\ v_8 \\ v_9 \end{pmatrix}, \, V = \begin{pmatrix} V_1 \\ V_2 \\ V_3 \\ V_4 \\ V_5 \\ V_6 \\ V_7 \\ V_8 \\ V_9 \end{pmatrix}, \, \omega_h = \begin{pmatrix} 0 \\ 0 \\ 0 \\ \omega_2 \\ \omega_2 \\ 0 \\ 0 \\ \omega_6 \\ \omega_7 \end{pmatrix}, \, \omega_r = \omega - \omega_h, \, V = -diag(v)\omega_r \tag{27}$$

Consequently, the viscous losses for the system of Equation (15) are given by

$$V_D = diag(n_s)V = -diag(n_s) \, diag(v) \, \omega_r \tag{28}$$

Introduce vectors $T$ of external torques, $T_T$ of the known external torques, and matrix $e_T$ of transition to the vector $T$, then

$$T = \begin{pmatrix} T_1 \\ 0 \\ 0 \\ T_4 \\ T_5 \\ M_{68} \\ M_{79} \\ M_{86} \\ M_{97} \end{pmatrix}, \, T_T = \begin{pmatrix} T_1 \\ T_4 \\ T_5 \\ M_{86} \\ M_{97} \end{pmatrix}, \, e_T = \begin{pmatrix} 1 & 0 & 0 & 0 & 0 \\ 0 & 0 & 0 & 0 & 0 \\ 0 & 0 & 0 & 0 & 0 \\ 0 & 1 & 0 & 0 & 0 \\ 0 & 0 & 1 & 0 & 0 \\ 0 & 0 & 0 & -1 & 0 \\ 0 & 0 & 0 & 0 & -1 \\ 0 & 0 & 0 & 1 & 0 \\ 0 & 0 & 0 & 0 & 1 \end{pmatrix}, \, T = e_T T_T \tag{29}$$

Consequently, the right part of the Equation (15) system is represented in matrix form as

$$T_D = H_T T = H_T e_T T_T \tag{30}$$

Thus, the system of Equation (15) is written as

$$I_D \varepsilon + E_D M = T_D + V_D \tag{31}$$

Note that the clutch friction moments are split to be represented as external torques along with shafts' torques in the vector $T$.

### 3.2. Kinematic Constraints

In this section, the technique for automating the description of the absolute and relative kinematic constraints for each mechanical interaction is derived. In describing the ideal kinematic connections adopted in the sport differential design, two moments

occur regarding the continuity of links transmitting both absolute and relative kinematic parameters[14]. Introduce the following basic kinematic vectors

$$\boldsymbol{\omega} = \begin{pmatrix} \omega_1 \\ \vdots \\ \omega_n \end{pmatrix}, \ \boldsymbol{\varepsilon} = \frac{d\omega}{dt} = \begin{pmatrix} \varepsilon_1 \\ \vdots \\ \varepsilon_n \end{pmatrix} \tag{32}$$

where $n$ = number of rotating components (9). In particular, as shown in Figure 2, some of the elements (2, 3, 4, 5) have differential relations, and the relative angular speed $\omega_3$ describes the motion of the satellites 3. Thus, for all couples (columns) of the matrix $\boldsymbol{L}$ Equation (1), except for containing the element 3, the gear ratios are defined as

$$i_{kl} = \omega_k / \omega_l \tag{33}$$

where $k = \{1, 2, 2, 4, 5\}$, $l = \{2, 6, 7, 8, 9\}$ = the positions of the couples' elements transmitting the absolute angular speeds.

In this case, the continuity equation of a kinematic connection can be written in general form as

$$\omega_k - i_{kl}\omega_l = 0, \ \varepsilon_k - i_{kl}\varepsilon_l = 0 \tag{34}$$

In the case of differential dependency between links, this yields

$$i_{kl} = i_{kl}^{(p)} = (\omega_k - \omega_p) / (\omega_l - \omega_p) \tag{35}$$

where $k = \{4, 5\}$, $l = \{5, 4\}$—the positions of the couple's elements transmitting the relative angular speeds, and $p = \{2, 2\}$ = positions of the stopped carrier.

The general remark regarding the sign of a gear ratio can be made, assuming that it is determined depending on the rotational directions of the corresponding parts. Thus, if both linked parts provide the same counterclockwise and clockwise rotations, their gear ratio is to be considered as positive regardless of whether the engagement is external or internal. Conversely, a change of direction means a negative gear ratio. For example, couple 3–5 (Figure 2) provides the rotation of satellites 3 counterclockwise if the side gear 5 also rotates counterclockwise. However, couple 3–4, which is symmetrical to it, provides the clockwise rotation of satellite 3 if the side gear 4 rotates counterclockwise, which corresponds to a negative gear ratio. Note that with this approach, the signs of the mechanism's force and kinematic gear ratios coincide using this approach.

As known, the differential mechanism provides two degrees of freedom, and at the same time, according to Figure 2, three couples of details are involved. It is assumed that the ratios $i_{34}$ and $i_{35}$ correspond to when carrier 2 is stopped. Differential constraints are described by the Willis formula, which gives the following statements for the case of power distribution

$$i_{45}^{(2)} = \frac{\omega_4 - \omega_2}{\omega_5 - \omega_2} = \frac{\omega_4^{(2)}}{\omega_5^{(2)}} = \frac{\omega_4^{(2)}}{\omega_3} \frac{\omega_3}{\omega_5^{(2)}} = -\frac{z_3}{z_4} \frac{z_5}{z_3} = \frac{i_{43}^{(2)}}{i_{53}^{(2)}} = \frac{i_{35}}{i_{34}} \tag{36}$$

where $z_3$ and $z_5$—teeth numbers of corresponding elements. The relative ratios between side gears and satellites are determined as follows

$$i_{43}^{(2)} = \frac{\omega_4 - \omega_2}{\omega_3} = \frac{\omega_4^{(2)}}{\omega_3} = \frac{1}{i_{34}}, \ i_{53}^{(2)} = \frac{\omega_5 - \omega_2}{\omega_3} = \frac{\omega_5^{(2)}}{\omega_3} = \frac{1}{i_{35}} \tag{37}$$

Equations (36) and (37) give the relative kinematic expressions for differential links as

$$(\omega_4 - \omega_2)i_{34} - (\omega_5 - \omega_2)i_{35} = 0, \ \omega_3 - (\omega_4 - \omega_2)i_{34} = 0, \ \omega_3 - (\omega_5 - \omega_2)i_{35} = 0 \tag{38}$$

Thus, the resulting system of the algebraic equations of kinematic connections, using the derivatives of Equations (37) and (38), is

$$
\begin{cases}
\varepsilon_1 - i_{12}\varepsilon_2 = 0 \\
(\varepsilon_4 - \varepsilon_2)i_{34} - (\varepsilon_5 - \varepsilon_2)i_{35} = 0 \\
\varepsilon_2 - i_{26}\varepsilon_6 = 0 \\
\varepsilon_2 - i_{27}\varepsilon_7 = 0 \\
\varepsilon_3 - (\varepsilon_4 - \varepsilon_2)i_{34} = 0 \\
\varepsilon_3 - (\varepsilon_5 - \varepsilon_2)i_{35} = 0 \\
\varepsilon_4 - i_{48}\varepsilon_8 = 0 \\
\varepsilon_5 - i_{59}\varepsilon_9 = 0
\end{cases}
\tag{39}
$$

This system can be decomposed considering vector $i$ and matrix $L$ in Equations (1) and (2)

$$
\boldsymbol{\varepsilon}_{in} = \begin{pmatrix} \varepsilon_1 \\ \varepsilon_2 \\ \varepsilon_2 \\ \varepsilon_2 \\ \varepsilon_3 \\ \varepsilon_3 \\ \varepsilon_4 \\ \varepsilon_5 \end{pmatrix}, \ \boldsymbol{\varepsilon}_{out} = \begin{pmatrix} \varepsilon_2 \\ \varepsilon_3 \\ \varepsilon_6 \\ \varepsilon_7 \\ \varepsilon_4 \\ \varepsilon_5 \\ \varepsilon_8 \\ \varepsilon_9 \end{pmatrix} - \begin{pmatrix} 0 \\ 0 \\ 0 \\ 0 \\ \varepsilon_2 \\ \varepsilon_2 \\ 0 \\ 0 \end{pmatrix}, \ \boldsymbol{\varepsilon}_{in} = e_{in}\boldsymbol{\varepsilon}, \ \boldsymbol{\varepsilon}_{out} = e_{out}\boldsymbol{\varepsilon} \tag{40}
$$

The matrices $e_{in}$ and $e_{out}$ can be obtained based on rows 1 and 2 of the matrix $L$, respectively. For this, two matrices are logically compared. One of which is obtained by repeating an elements' column-vector several times equal to the number of connections. The second is obtained by copying a row of matrix $L$ (upper for $e_{in}$ or lower for $e_{out}$) several times equal to the number of elements being considered ($n$ in this case). The result will give logical matrices containing 0 and 1. Note that link 2–3 in the matrix $L$ describing the interaction between carrier 2 and satellites 3 is differential, and therefore the corresponding row in the vectors $\boldsymbol{\varepsilon}_{in}$ and $\boldsymbol{\varepsilon}_{out}$ must be zeroed and links 3–4 and 3–5 represented by relative angular accelerations following Equation (38). To do this, introduce the matrix $e_d$, which ensures the replacement of the conjugation of the carrier and satellites rotating in different planes with a differential connection by subtracting the angular accelerations for the side gears and satellites relative to the satellite. Thus, Equation (39) in matrix form is

$$
e_\varepsilon = e_d(e_{in} - diag(i)e_{out}), \ e_\varepsilon\boldsymbol{\varepsilon} = z \tag{41}
$$

where $z$ = zero column-vector with the length corresponding to the number of kinematic links.

On the other hand, the differential is characterized by two degrees of freedom. That is, the vector $\boldsymbol{\varepsilon}$ of angular accelerations can be expressed through the vector $\boldsymbol{\varepsilon}_D$ containing only the angular accelerations of the side gears

$$
\boldsymbol{\varepsilon}_D = \begin{pmatrix} \varepsilon_4 \\ \varepsilon_5 \end{pmatrix}, \ \boldsymbol{\varepsilon}_D = \begin{pmatrix} \varepsilon_4 \\ \varepsilon_5 \end{pmatrix}, \ \boldsymbol{\varepsilon} = E_\varepsilon\boldsymbol{\varepsilon}_D \tag{42}
$$

where

$$E_\varepsilon = \frac{1}{(i_{34} - i_{35})} \begin{pmatrix} i_{12}i_{34} & -i_{12}i_{35} \\ i_{34} & -i_{35} \\ -i_{35}i_{34} & i_{35}i_{34} \\ (i_{34} - i_{35}) & 0 \\ 0 & (i_{34} - i_{35}) \\ i_{34}/i_{26} & -i_{35}/i_{26} \\ i_{34}/i_{27} & -i_{35}/i_{27} \\ (i_{34} - i_{35})/i_{48} & 0 \\ 0 & (i_{34} - i_{35})/i_{59} \end{pmatrix} \tag{43}$$

Thus, substituting Equation (43) into Equation (41) helps to reduce the number of unknown kinematic variables. Thus,

$$e_\varepsilon = e_d(e_{in} - diag(i)e_{out}), \; e_\varepsilon E_\varepsilon \varepsilon_D = z \tag{44}$$

*3.3. Gearing Conditions*

The technique for reducing several unknown internal moments along with the approach for automating the determination of moments' signs are represented in this section. Consider the force factors emerging in conjugated components[11]. In the general case, the sum of the moments at a node can be expressed as

$$M_k i_{kl} + M_l = 0 \tag{45}$$

where $k$, $l$ = indexes of conjugated details and $i_{kl}$ = ratio.

Note that in the general case, $i_{kl}$ can be either positive or negative depending on whether the connection changes the moment sign. Thus, for example, for an internal gearing pair, the directions of positive moments coincide. Therefore, the gear ratio is positive, which determines the opposite signs of $M_l$ and $M_k$ as the driving and reaction moments in a node.

Assuming that a part of mechanical energy is lost while transmitting the moment, for the cases of direct and reverse power flow, it can be, respectively, written as

$$M_{kl}i_{kl}\eta_{kl} + M_{lk} = 0, \; M_{kl}i_{kl} + M_{lk}\eta_{lk} = 0 \tag{46}$$

where $\eta_{kl}$ and $\eta_{lk}$ = direct and reverse gearing efficiencies, respectively (it may be assumed that $\eta_{kl} = \eta_{lk}$).

Let $\boldsymbol{\eta}_G$ be the vectors of gearing efficiencies for exceptionally differential's details, $\boldsymbol{\eta}_C$ be the vector of gearing efficiencies for clutches' details, and $\boldsymbol{M}_D$ be the vector of unknown internal moments

$$\boldsymbol{\eta}_G^{(s)} = \begin{pmatrix} \eta_{12}^{(s)} \\ \eta_{23}^{(s)} \\ 1 \\ 1 \\ \eta_{34}^{(s)} \\ \eta_{35}^{(s)} \\ 1 \\ 1 \end{pmatrix}, \boldsymbol{\eta}_G^{(c)} = \begin{pmatrix} \eta_{12}^{(c)} \\ \eta_{23}^{(c)} \\ 1 \\ 1 \\ \eta_{34}^{(c)} \\ \eta_{35}^{(c)} \\ 1 \\ 1 \end{pmatrix}, \boldsymbol{\eta}_C^{(s)} = \begin{pmatrix} 1 \\ 1 \\ \eta_{26}^{(s)} \\ \eta_{27}^{(s)} \\ 1 \\ 1 \\ \eta_{48}^{(s)} \\ \eta_{59}^{(s)} \end{pmatrix}, \boldsymbol{\eta}_C^{(c)} = \begin{pmatrix} 1 \\ 1 \\ \eta_{26}^{(c)} \\ \eta_{27}^{(c)} \\ 1 \\ 1 \\ \eta_{48}^{(c)} \\ \eta_{95}^{(c)} \end{pmatrix}, \boldsymbol{M}_D = \begin{pmatrix} M_{21} \\ M_{32} \\ M_{62} \\ M_{72} \\ M_{43} \\ M_{53} \\ M_{84} \\ M_{95} \end{pmatrix} \tag{47}$$

where $s$, $c$ = upper indexes for determining the possible state and converse state efficiencies, respectively, depending on whether the power is being transmitted according to the order the nodes in the matrices $\boldsymbol{L}$ and $\boldsymbol{C}$ or vice versa.

Thus, the systems of moments' equilibrium for the generalized case of direct or reverse power flow is

$$\begin{cases} M_{21}\eta_{12}^{(c)} + M_{12}\eta_{12}^{(s)}i_{12} = 0 \\ M_{32}n_s\eta_{23}^{(c)} + M_{23}\eta_{23}^{(s)}i_{23} = 0 \\ M_{62}\eta_{26}^{(c)} + M_{26}\eta_{26}^{(s)}i_{26} = 0 \\ M_{72}\eta_{27}^{(c)} + M_{27}\eta_{27}^{(s)}i_{27} = 0 \\ M_{43}\eta_{34}^{(c)} + M_{34}\eta_{34}^{(s)}i_{34}n_s = 0 \\ M_{53}\eta_{35}^{(c)} + M_{35}\eta_{35}^{(s)}i_{35}n_s = 0 \\ M_{84}\eta_{48}^{(c)} + M_{48}\eta_{48}^{(s)}i_{48} = 0 \\ M_{95}\eta_{59}^{(c)} + M_{59}\eta_{59}^{(s)}i_{59} = 0 \end{cases} \tag{48}$$

Note that the system of Equation (48) depends on the states of power flows passing through the differential and through the clutches. For each pair of $k = \{1, 2, 3, 3\}$ and $l = \{2, 3, 4, 5\}$, possible states are switched simultaneously since these parts are being connected in the whole mechanism. Thus,

$$\eta_G^{(s)} \in \left\{ \eta_G^{(d)}, \quad \eta_G^{(r)} \right\}, \eta_G^{(c)} \in \left\{ \eta_G^{(r)}, \quad \eta_G^{(d)} \right\},$$
$$\eta_G^{(d)} = \eta_G \Rightarrow \eta_G^{(r)} = \{1\}, \eta_G^{(r)} = \eta_G \Rightarrow \eta_G^{(d)} = \{1\} \tag{49}$$

where $\eta_G$ = gearing efficiency vector with $\eta_{kl}$, and $\{1\}$ = vector of ones with the same size as $\eta_G$.

Unlike those mentioned above, the clutches' elements can be active and passive depending on the friction moments $M_{86}$ and $M_{97}$ regardless of the basic mechanism state. A clutch may be withdrawing some torque from the carrier and returning it back. For each pair of $k = \{6, 7, 8, 9\}$ and $l = \{2, 2, 4, 5\}$, the sign of the difference in moments may determine the state since an angular velocity is the same at both sides of an element. Thus,

$$|M_{kl}| - |M_f| > 0 \tag{50}$$

where the friction moment $M_f = \{M_{86}, M_{97}\}$ for corresponding indexes $k$ and $l$, respectively.

If the inequality of Equation (50) for a clutch component is satisfied, its mode will be "direct" and vice versa. This implies the following assertions, where the upper indexes denote conditionally direct (*d*) and reverse (*r*) states. Then,

$$\eta_{kl}^{(s)} \in \left\{ \eta_{kl}^{(d)}, \quad \eta_{kl}^{(r)} \right\}, \eta_{kl}^{(c)} \in \left\{ \eta_{kl}^{(d)}, \quad \eta_{kl}^{(r)} \right\},$$
$$\eta_{kl}^{(d)} = \eta_{kl} \Rightarrow \eta_{kl}^{(r)} = 1, \eta_{kl}^{(d)} = \eta_{kl} \Rightarrow \eta_{kl}^{(r)} = 1 \tag{51}$$

The pairwise use of the moments in Equation (48) and their sequential pair arrangement in the vector $M$ makes it possible to separate the parts of action and reaction moments through the matrices $e_r$ at gear ratios and $e_m$ for output loads of dimension $m \times 2m$. Thus, in the matrix $e_r$, the elements corresponding to the mesh nodes based on each row and each second column, starting from the first, are equal to 1, while the other elements equal 0. The same procedure for forming the matrix $e_m$ but starting from the second column. Thus, the system of Equation (48) is represented in matrix form as follows:

$$e_M M = z \tag{52}$$

where for the generalized state of the power flows,

$$e_M = \left( H_G^{(s)} diag(i) e_r + H_G^{(c)} e_m \right) diag(N_s) \tag{53}$$

On the other hand, based on the system of Equation (47), all the moments of vector $M$ can be expressed by the elements of vector $M_D$. Considering Equation (24), it can be derived for the generalized case as

$$E_M = (diag(N_D))^{-1} \left( e_m - \left( H_G^{(s)} diag(i) \right)^{-1} H_G^{(c)} e_r \right)^T \tag{54}$$

where $N_D$ may be obtained by replacing elements 3 and 4 in the vector $N_s$ with $1/n_s$ and $1$, respectively. Then,

$$M = E_M M_D, \quad e_M E_M M_D = z \tag{55}$$

Another step is regarding replacing $M_{32}$ with an expression based on $M_{43}$ and $M_{53}$. An additional condition can be formed because the driving moments of the differential carrier and satellites lie in different planes, as shown in Figure 5e. Then, for a satellite,

$$P_{32} + P_{34} + P_{35} = 0 \tag{56}$$

Multiply by the radius $R_{23}$ of the force transmission through the satellite's axle and by the number of satellites, then

$$n_s P_{32} R_{23} + n_s P_{34} R_{23} + n_s P_{35} R_{23} = 0 \tag{57}$$

or

$$n_s M_{32} + n_s P_{34} R_3 \frac{R_{23}}{R_3} + n_s P_{35} R_3 \frac{R_{23}}{R_3} = 0,$$
$$n_s M_{32} + n_s M_{34} i_{34} u_{24} + n_s M_{35} i_{35} u_{25} = 0 \tag{58}$$

where $u_{24} = R_{23}/R_4$, $u_{25} = R_{23}/R_5$ = ratio of passing forces between the radii of satellites' axles and gearings (may be accepted that $u_{24} = u_{25} = 1$).

Using Equation (48), Equation (58) is rewritten in the generalized form as

$$M_{32} \frac{\eta_{23}^{(c)} n_s}{\eta_{23}^{(s)} i_{23}} = M_{43} \frac{\eta_{34}^{(c)}}{\eta_{34}^{(s)}} + M_{53} \frac{\eta_{35}^{(c)}}{\eta_{35}^{(s)}} \tag{59}$$

Let

$$b_{43} = \frac{\eta_{34}^{(c)}}{\eta_{34}^{(s)}} \frac{\eta_{23}^{(s)} i_{23}}{\eta_{23}^{(c)} n_s}, \quad b_{53} = \frac{\eta_{35}^{(c)}}{\eta_{35}^{(s)}} \frac{\eta_{23}^{(s)} i_{23}}{\eta_{23}^{(c)} n_s} \tag{60}$$

Then,

$$M_{32} = M_{43} b_{43} + M_{53} b_{53} \tag{61}$$

The latter excludes the redundant variable $M_{32}$ from $M_D$ and reduces the unknown variables.

Denote transition matrix $S$ and vector $M_U$ of independent internal moments. Then

$$S = \begin{pmatrix} 1 & 0 & 0 & 0 & 0 & 0 & 0 \\ 0 & 0 & 0 & b_{43} & b_{53} & 0 & 0 \\ 0 & 1 & 0 & 0 & 0 & 0 & 0 \\ 0 & 0 & 1 & 0 & 0 & 0 & 0 \\ 0 & 0 & 0 & 1 & 0 & 0 & 0 \\ 0 & 0 & 0 & 0 & 1 & 0 & 0 \\ 0 & 0 & 0 & 0 & 0 & 1 & 0 \\ 0 & 0 & 0 & 0 & 0 & 0 & 1 \end{pmatrix}, \quad M_U = \begin{pmatrix} M_{21} \\ M_{62} \\ M_{72} \\ M_{43} \\ M_{53} \\ M_{84} \\ M_{95} \end{pmatrix}, \quad M_D = S M_U \tag{62}$$

### 3.4. Friction Clutch

The additional delivery of a power flow in this DM design is carried out by the external control according to the need (dashed lines in Figure 3) and corresponds to the interactions of nodes specified in the matrix $C$ of Equation (1) [4]. The formation of driving moments in friction clutches is schematically shown in Figure 5f–h.

#### 3.4.1. Friction Torque

It is evident that in the vector $T$ from Equation (29), $M_{68} = -M_{86}$ and $M_{79} = -M_{97}$, which allows considering only the driven parts. Frictional moments in clutches are functions of the design parameters and normal forces $N_{10}$, $N_{11}$. That is,

$$M_{kl} = f_{kl} N_p R_p n_{kl} \tag{63}$$

where $M_{kl}$ = driving frictional moment, where $k$ = {8, 9} and $l$ = {6, 7} = indexes denoting frictional elements, $N_p$ = normal force, $p$ = {10, 11} = indexes denoting pressing elements, $R_p$ = average friction radius, $f_{kl}$ = friction factor, and $n_{kl}$ = number of friction surfaces.

The hyperbolic tangent function can be accepted as a frictional model that ensures automatic changing of the sign when the clutch slides. Additionally, this model is sufficient since the design does not provide strict conditions for locking clutches, as it is required, for example, in the automatic gearbox. Moreover, the torque transmitted to a wheel should preferably be proportional to the pressure in a hydraulic cylinder, which makes the control predictable and stable.

$$f_{kl} = \mu_{kl} \tanh(c_{kl} \Delta \omega_{kl}), \ \Delta \omega_{kl} = (\omega_l - \omega_k) \tag{64}$$

where $\mu_{kl}$ = the module value of friction coefficient, and $c_{kl}$ = the intensity coefficient.

#### 3.4.2. Lockup Condition

The algorithm for determining the current locking compression force is proposed here for use in the simulation model. Despite the possibility of conditionally unlimited increasing the normal force $N_p$ in Equation (32), the frictional moment is restricted by the critical value $N_{pc}$, leading to locking a clutch. At the same time, an $N_{pc}$ value does not remain constant, which is consistent with external loading changes. In this regard, it is necessary to provide an algorithm for setting the locking mode and limiting the friction torque. For this, introduce a threshold zone $\pm \Delta \omega_c$ in the vicinity of $\Delta \omega_{kl} = 0$, where the mode is equivalent to the lock state, and the sliding is practically absent. The current value of the clutch compression force $N_p$ can result in two states:

$$|\Delta \omega_{kl}| > \Delta \omega_c, \ |\Delta \omega_{kl}| \leq \Delta \omega_c \tag{65}$$

If the absolute value of the relative angular speed does not exceed a threshold value, the $N_p$ value is stored in the memory as $N_{p(-1)}$. In the next step, the new value $N_{p(+1)}$ can be compared with the previous one for the minimum, as follows

$$N_p = min \left\{ \begin{array}{cc} N_{p(-1)} & N_{p(+1)} \end{array} \right\} \tag{66}$$

Note that an $N_p$ value is not related to a friction moment sign. This point stipulates only positive values of $N_p$. The friction moment in this state is defined as

$$M_{kl} = \mu_{kl} N_p R_p n_{kl} \mathrm{sgn}(\Delta \omega_{kl}) \tag{67}$$

When $\Delta \omega_{kl}$ exceeds a threshold value, the $N_p$ current value is taken as the basic one and cycles are repeated.

### 3.4.3. Piston Dynamics

To reflect the elastic and stiff properties of clutch discs as parts of the clutch pack compression physics, consider the translational motion of pistons compressing the clutches, as shown in Figure 5h. The system of equations can be represented as

$$\begin{cases} dx_p/dt = s_p \\ m_p a_p = P_p + N'_p \end{cases} \tag{68}$$

where $m_p$ = piston mass, $a_p$ = piston acceleration, $s_p$ = piston velocity, $x_p$ = piston translational displacement, $P_p$ = resulting piston force, $N'_p$ = the force of combined normal reaction, and $p$ = the index of translational element ($p$ = 10, 11).

Denote

$$P_p = p_p A_p + F_p + S_p \tag{69}$$

where $p_p$ = the pressure in the cylinder $p$, $A_p$ = the piston area, $F_p$ = the resistance force of interaction between the cylinder and the cuff, and $S_p$ = elastic force of return springs.

The components of Equation (69) may be found by the following formulas, considering $p$ = 10, 11:

$$A_p = \frac{\pi}{4} \left( d_{pe}^2 - d_{pi}^2 \right) \tag{70}$$

where $d_{pe}$, $d_{pi}$ = piston external and internal diameters, respectively.

$$S_p = -k_{sp} \left( x_p + \Delta_{p0} \right) \tag{71}$$

where $k_{sp}$ = spring stiffness and $\Delta_{p0}$ = initial deformation.

$$F_p = - \left( p_p h \pi \left( d_{pe} + d_{pi} \right) f_p + v_p s_p \right), \quad f_p = \mu_p \tanh \left( c_p s_p \right) \tag{72}$$

where $h$ = cuff width, $f_p$ = friction factor, $v_p$ = viscous coefficient, $\mu_p$ = module value of the translational friction coefficient, and $c_p$ = the intensity coefficient.

The simplest way to form the force $N'_p$ consists of setting piecewise linear functions depending on the piston stroke. The $N'_p$ reaction ensures the piston hard stop in the boundary positions, while the $N_p$ component is the clutch compressing force.

$$N'_p = -k_{p1} x_p E_H (-x_p) - N_p,$$
$$N_p = k_{p2} \left( x_p - x_{p\Delta} \right) \left( E_H \left( x_p - x_{p\Delta} \right) - E_H \left( x_p - x_{h\Delta} \right) \right) + P_{po} E_H \left( x_p - x_{h\Delta} \right) \tag{73}$$

where $k_{p1}$, $k_{p2}$ = the stiffness of elastic forces in the piston boundary positions, $x_{p\Delta}$ = clearance for ensuring the complete clutch disengagement, $x_{h\Delta}$ = piston stroke corresponding to the hard-stop, $P_{po}$ = value of the $P_p$ force evaluated at the preceding time-step, and $E_H$ = Heaviside step-function.

### 3.5. Matrix Form

The matrix form is the most efficient for such equations [8]. Therefore, the three different types of equation systems are combined to automate the allocation of efficiencies, ratios, unknown moments, and their signs and reduce the size of the total equation system and the number of variables. Thus, summarizing all the preceding, the systems of Equations (15), (39), and (48), considering the equations of the previous section to form the vector $\boldsymbol{T_D}$ from Equation (30), can be rewritten in matrix form as

$$\begin{cases} \boldsymbol{I_D} \boldsymbol{E}_\varepsilon \boldsymbol{\varepsilon_D} + \boldsymbol{E_D} \boldsymbol{E_M} \boldsymbol{M_D} = \boldsymbol{T_D} + \boldsymbol{V_D} \\ \boldsymbol{e}_\varepsilon \boldsymbol{E}_\varepsilon \boldsymbol{\varepsilon_D} = \boldsymbol{z_m} \\ \boldsymbol{e_M} \boldsymbol{E_M} \boldsymbol{M_D} = \boldsymbol{z_m} \end{cases} \tag{74}$$

where $\boldsymbol{z_m}$ = vector of zeros with dimension $m \times 1$.

Denote the following matrices for Equation (74):

$$D = \begin{pmatrix} I_D & E_D \\ e_\varepsilon & Z_{m,2m} \\ Z_{m,n} & e_M \end{pmatrix}, F = \begin{pmatrix} E_\varepsilon & Z_{n,m} \\ Z_{2m,2} & E_M \end{pmatrix}, G = \begin{pmatrix} E_{2,2} & Z_{2,m-1} \\ Z_{m,2} & S \end{pmatrix},$$

$$R = \begin{pmatrix} T_D + V_D \\ z_m \\ z_m \end{pmatrix}$$

(75)

where $Z_{q,r}$ = matrix of zeros with dimension $q \times r$, and $E_{2,2}$ = identity matrix $2 \times 2$.

According to the properties of Equations (44) and (55), the kinematic and gearing equations in Equation (74) become zeros and, therefore, may be reduced by the rectangular identity matrix $E_U$ with dimension $n \times n + 2m$. Consequently, the final matrix components are obtained as

$$x = \begin{pmatrix} \varepsilon_D \\ M_U \end{pmatrix}, E_U = E_{n,n+2m}, B = E_U R, A = E_U DFG, x = A^{-1}B$$

(76)

## 4. Simulation

### 4.1. Simulink Model

Based on the preceding theoretical expressions, a simulation model (Figure 6) of the sport differential mechanism and its control under external loads conditions was composed, validated, and tested for different cases using MATLAB software [19]. The blocks in Figure 6 are fully adjustable and configurable and can be used as a component of an intelligent transmission Simulink model for developing control drives and algorithms.

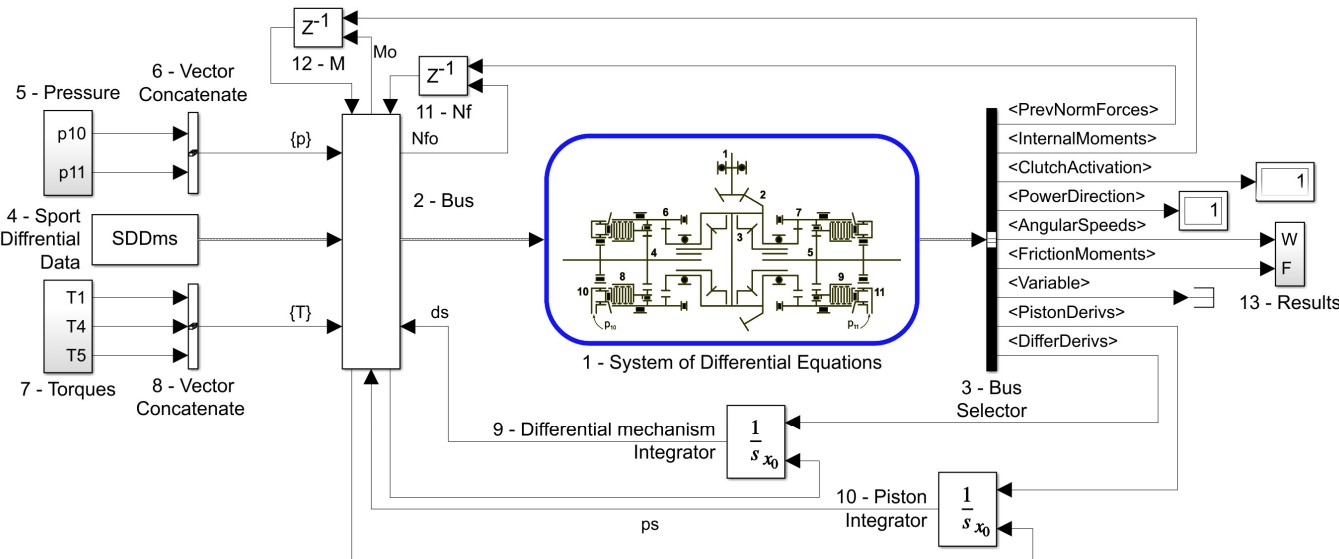

**Figure 6.** Simulink-model of the sport differential functioning.

The model's basic element is block *1 (System of differential Equations)*, which contains a description code of differential equations for translationally and rotationally moving design parts. The block has complex input and output ports transmitting information through buses. In block *2 (Bus)*, the required data about the model components' states are updated, and vectors of initial conditions for integrators and delay blocks are distributed.

Block *3 (Bus Selector)* expands the complex output by the corresponding signal names. Namely, signals denote: *<PrevNormForces>* is a row vector of compression forces in friction clutches from previous integration step; *<InternalMoments>* is a vector of internal moments used for detecting clutches' operating modes and states of transmitting elements; *<ClutchActivation>* is a row vector of frictional couplings 6–8 and 7–9 modes taking values 0—inactive and 1—activated; *<PowerDirection>* is the direction of the external power flow for the differential mechanism: 1 (direct) and -1 (reverse); *<AngularSpeeds>* is a column vector of angular velocities according to Equation (32); *<FrictionMoments>* is a row vector of frictional moments' values; *<Variable>* is a reserve variable for displaying any other data while testing the model; *<PistonDerivs>* are the derivatives of hydraulic (translational elements) clutches' states; *<DifferDerivs>* are the derivatives of the differential (rotational) components' states. All the needed information about the differential components' physical and geometric characteristics is collected in the *SDDms* structure and transmitted through block *4—Sport Differential Data*.

In block *5 (Pressure)*, the initial control signals for hydraulic cylinders 10 and 11 are formed and then combined in a row vector in block *6 (Vector Concatenate)*. Similarly, in block *7 (Torques)*, the external loads on the differential's shafts 1, 4, and 5 are generated to be combined into a column vector in block *8 (Vector Concatenate)*. The model uses two independent integrators: *9 (Differential Mechanism Integrator)* for the rotational motion equations and *10 (Piston Integrator)* for the equations of pistons' translational motion. Blocks *11 (Nf)* and *12 (M)* allow storing the vectors of compression forces and internal moments at the previous computing step in memory. Block *13 (Results)* accumulates the primary output data for analysis.

*4.2. Testing Differential Model Operability*

The model of DM must be tested first. It is expected that the distribution of angular speeds, in this case, corresponds to the concept of "least resistance" for a power flow. Thus, the most general test mode may be formed using periodical loads with the same phases but different amplitudes (Figure 7, External torque) on the mechanism shafts. The primary output information is the values of angular velocities of all the mechanism's rotating components. In turn, it can be stated that the shapes of the angular speeds' curves for shafts 1, 4, and 5 are in good coordination, reflecting trends of the corresponding external torques. The internal moments' curves also inherit the nature of the external ones with distinctive step shifts when the power flow direction is changed. The solution's periodicity remains, which indicates the model stability. The main point implies that an axle with lower loading torque tends to have a higher angular speed, typical for symmetrical frictionless differentials.

Note that the instantaneous switch in Figure 7 occurs at the time of mutual intersection of the external and internal loads in the nodes with zero values. The regime change means switching between equations with different efficiency vectors. From the point of view of zero loads, the computations of the dynamics equations for a given moment are the same. To eliminate this phenomenon, it is necessary to use efficiencies depending on torques and kinematics, which complicates the model and is insignificant for speed modes. Further, most models proposed in the literature do not use mode switching at all and often neglect mechanical efficiency.

Consider now an option of activating one of the differential's clutches (Figure 8) for the same external loading conditions as in Figure 7. Proceeding from the fact that a larger external resistance torque passes through axle 4, the pressure increase $p_{10}$ can be set while $p_{11}$ remains zero (Figure 8, Pressure). Only $\omega_4$ and $\omega_5$ angular speeds are reflected as the output. The pressure increase phases are accompanied by the friction torque $M_{86}$ ($M_{97} = 0$ because of $p_{11} = 0$). In the phase of positive values, this friction torque adds power to shaft 4 with a larger load, causing its angular speed $\omega_4$ to exceed the $\omega_5$ one, unlike the situation in Figure 7. The internal moments $M_{62}$ and $M_{84}$ appear in the opposite phase, revealing their possible leading/driven states when the clutch 6–8 is activated. Thus, in contrast to

the situation in Figure 7, the torque and speed output parameters are redistributed, and the angular speed of a more loaded shaft can exceed that of a less loaded shaft.

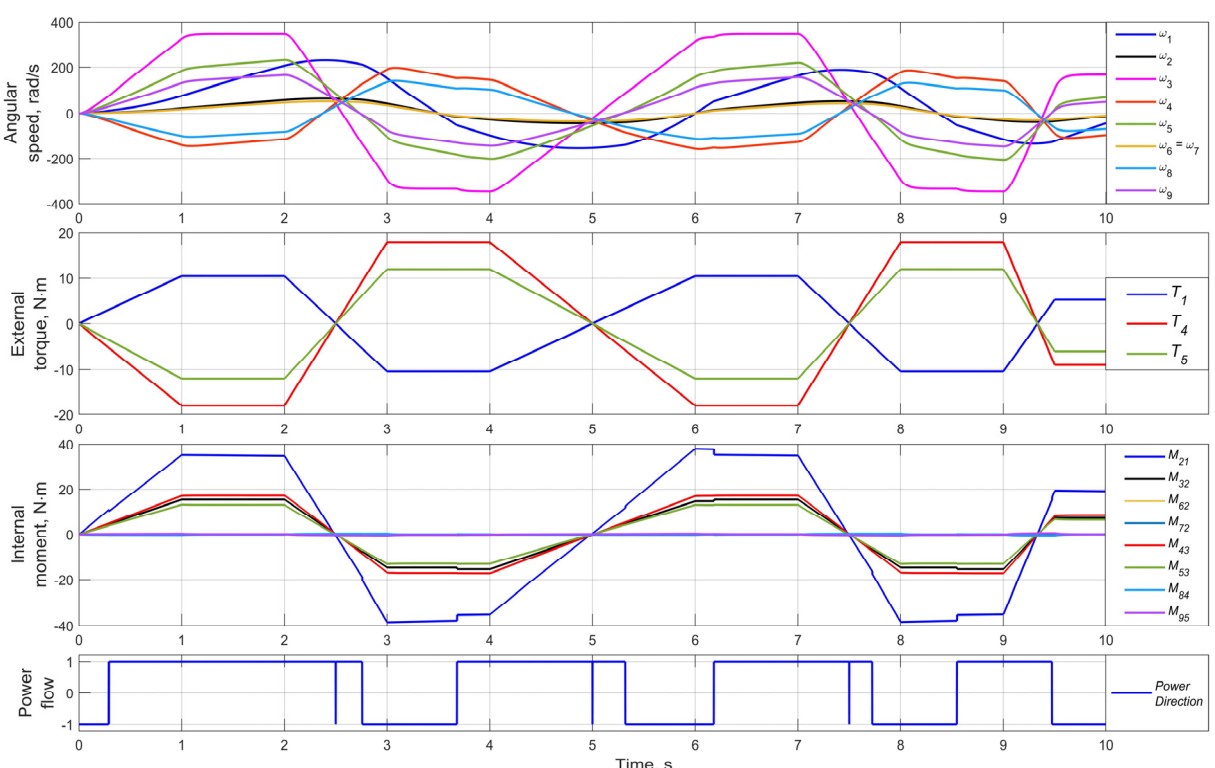

**Figure 7.** Modeling the mode of a conventional inter-axle differential.

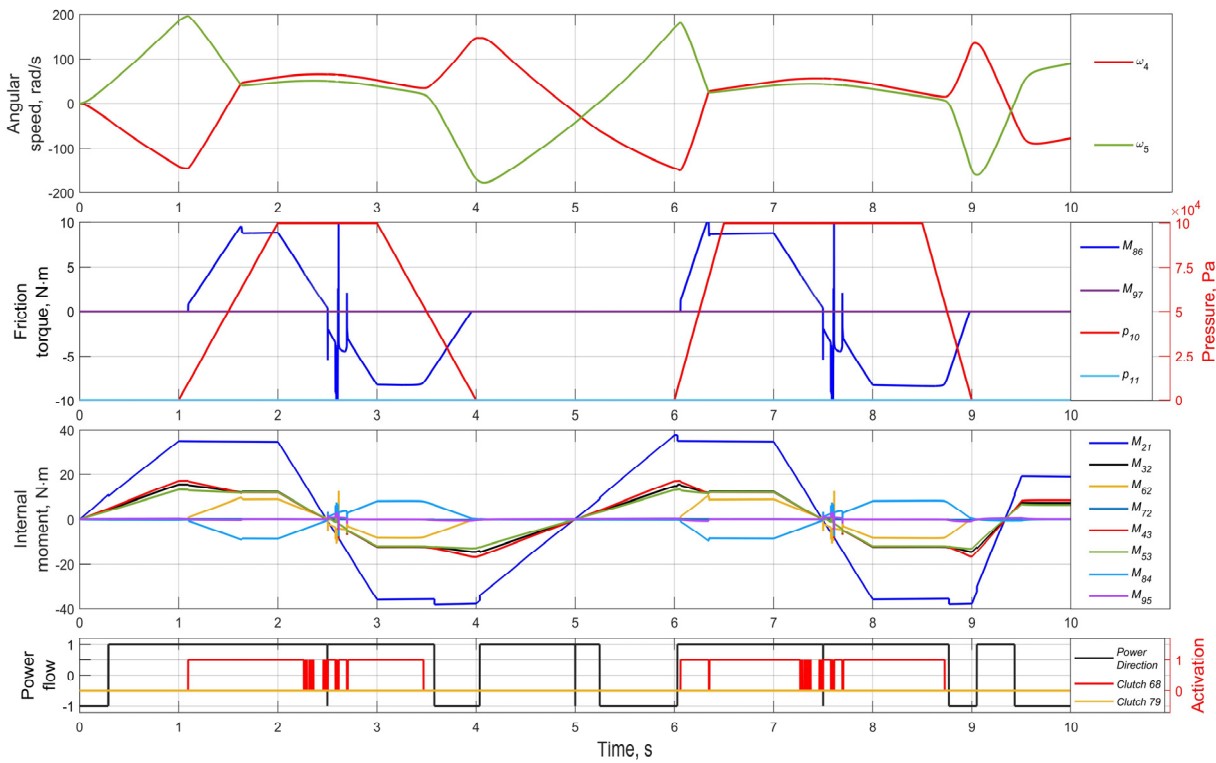

**Figure 8.** Modeling the mode of activating clutch 6–8.

Note that the spikes in Figure 8. are transient processes in friction clutches that inevitably accompany the changes of the load signs. The Simulink model is carefully tuned to minimize the influence of abrupt slip processes in clutch packs. The presence of dry friction in models always leads the differential equations to a "stiff" form that requires reducing the time increment to ensure the solution's stability.

### 4.3. Zero-Resistance Case

The case when one of the differential's output axles is unloaded (or the load tends to zero) is of interest. Such a situation typically occurs due to the lack of tire-road adhesion on one of the same axle wheels. In the case of a conventional symmetrical differential, all the power goes to drive the wheel with the worst adhesion conditions, and its angular speed tends to a maximum. In the case of an active limited-slip differential, a part of the carrier's torque may be transmitted to a lagging axle, maintaining the differences in the angular speeds. These variants' properties can be compared with the example in Figure 9. Thus, axle 5 is unloaded, and the torques on shafts 1 and 4 are constant.

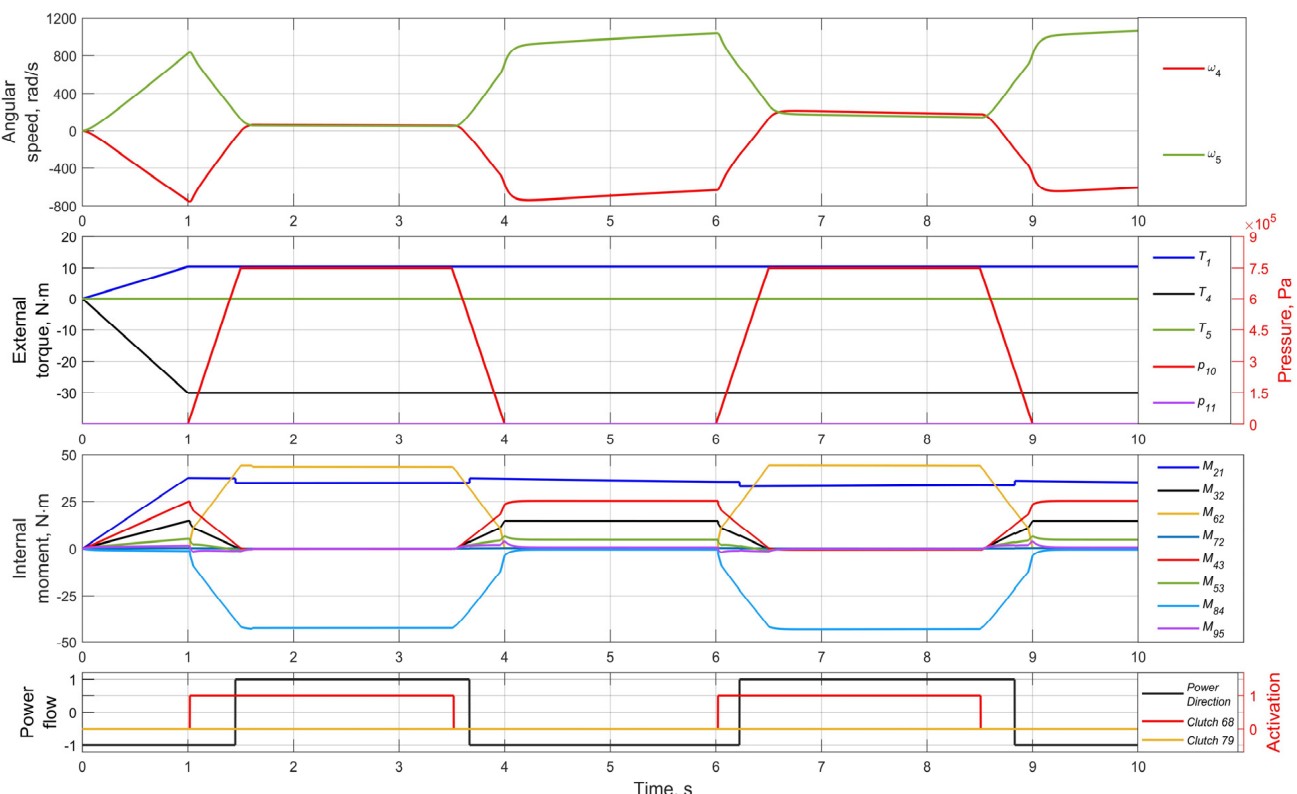

**Figure 9.** Simulation of transmitting all the torque through one output axle.

As seen, outside the time moments of activating the clutch 6–8, the angular speed of the unloaded axis 5 rapidly increases, and axis 4 rotates in the opposite direction under the influence of a negative load, which corresponds to the operational mode of a conventional differential. However, in the periods 1.5–3.5 s and 6.5–8.5 s of stable pressure in the clutch's 6–8 cylinder, the angular speed $\omega_4$ of the loaded shaft even slightly exceeds the angular speed $\omega_5$. At the same time, the moment $M_{43}$ value of the side gear 4 falls to a minimum, while the moments $M_{62}$ and $M_{84}$ reach their maximum modules. This situation corresponds to transmitting the maximum torque through one output shaft. Thus, the wheels' angular speeds are synchronized to ensure maximum vehicle passability.

### 4.4. Alternating Activation

The following example shows the alternating actuation of the friction clutches. This option can be used when correcting the trajectory under conditions of a motion track with a variable sign periodic curvature. Suppose the resisting moments to be changed periodically, and the driving moment is constant, as shown in Figure 10. At the same time, the load moments are alternately changed to larger/smaller amplitude values. A conventional differential design would result in a higher angular speed on the shaft with less load and vice versa. The activation of the friction clutches is organized in such a way to match the increase in axles' loads. As an immediate result, only the angular velocities $\omega_4$ and $\omega_5$ are taken accompanied by piston strokes $x_{p10}$, $x_{p11}$. As shown seen, the activation of the corresponding clutches causes the angular speed to rise for a semi-axle under higher load conditions, which allows more torque to be transmitted to a wheel with better adhesion conditions, ensuring a greater reaction value (traction force).

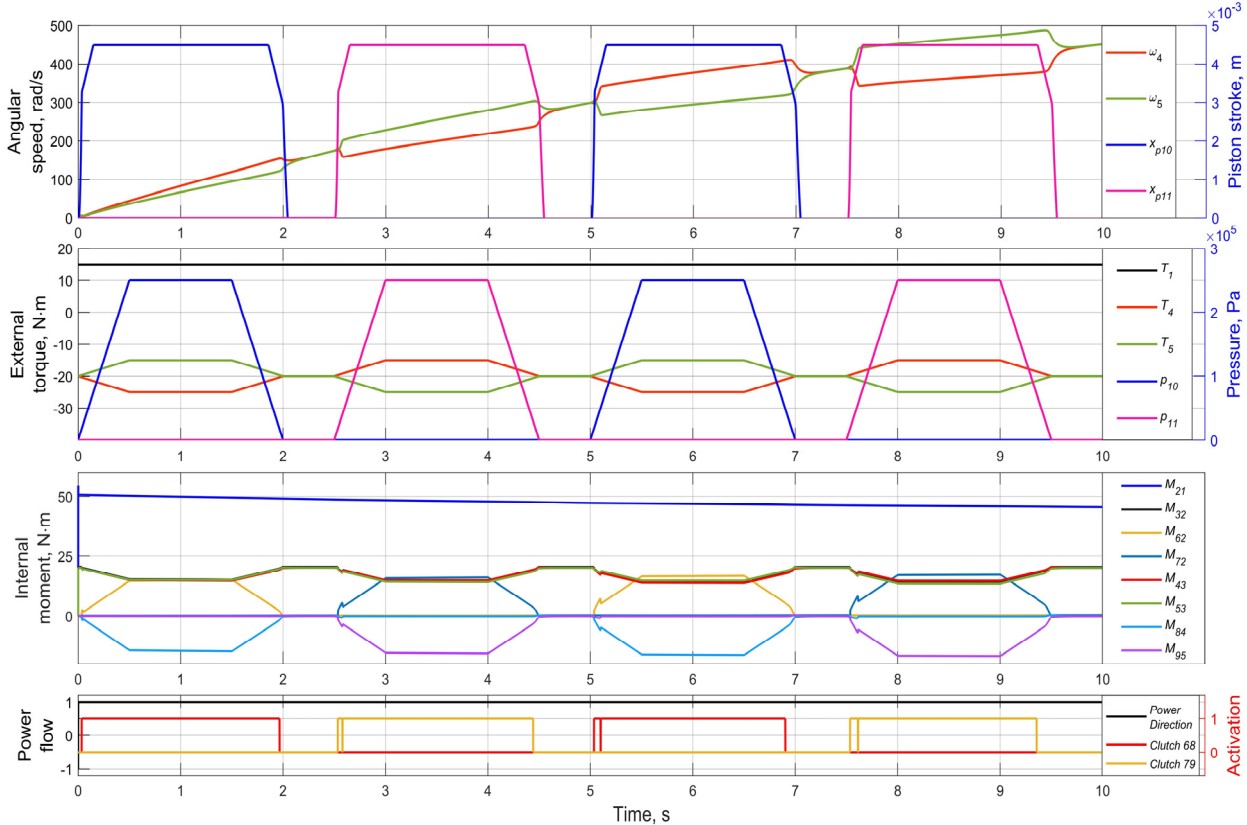

**Figure 10.** Simulation of alternating activation of friction clutches.

### 4.5. Analysis of Results

A few comments are noted about the data used and their impact on the simulation results. First, the question concerns the moments of friction forces and their influence on the solution accuracy. Varying the data has shown that the friction torque's increased sensitivity relative to the cylinder pressure negatively affects the slip smoothness. Thus, the variant of "hard" friction disks leads to the appearance of jerks and solution instability. The decreased sensitivity requires a larger pressure and may lead to a delay in the response time of a controlling drive. Thus, the compromise must meet the demands of operating properties and design compactness. Note that a decrease in the friction coefficient for clutches and an increase in the number of frictional pairs positively affect the smoothing of the torque passing from the differential's carrier. This issue can also be facilitated by introducing a piecewise constant compressing stiffness of a clutch package up to a deformation limit.

Thus, the adjustment and tuning of this design simulation model compose a separate task for adapting the model functioning to a specific range of loads and vehicle motion modes.

## 5. Conclusions

All the needed stages to achieve the results showed their adequacy, consistency, and coordination between each other. Based on this study, two main conclusions are made.

1.  All the simulations carried out with different sets of initial conditions confirm the model's efficiency in transmitting a larger torque to an axle with higher resistance and equalizing the angular speeds of the output shafts. This is unlike the working principle of functioning the conventional (open) and passive limited-slip differentials. The angular speed growth on a shaft with higher resistance leads to an increase in slip (up to a critical) and, as a rule, to a rise in traction force on a wheel, which contributes to the appearance of an additional turning moment relative to the vertical vehicle axis (Figure 1). At the same time, the friction clutch usage can be practical not only in the traction mode but also in the driven one, when due to the activation of the outer wheel clutch, the negative longitudinal reaction decreases. This fosters the wide use of various algorithms for controlling the sport differential to stabilize/align the vehicle trajectory. In addition, the possibility of transmitting all the torque to one of the output axles was demonstrated to maintain the vehicle's passability in the conditions of limited road-tire adhesion. Thus, the proposed sport differential model can be used for simulating the active control vehicle transmissions.

2.  The paper has proposed an alternative method for obtaining differential equations that describe the dynamics of rotational mechanical systems. As demonstrated, the main idea consisted of decomposing a mechanical system onto elementary components with the independent formation of three types of equations: dynamics, kinematic constraints, and force interactions. All the internal efforts' signs are determined automatically. The developed mathematical apparatus effectively reduces the total number of equations for compactness and lowers the simulation time. Thus, the approach corresponds to the modern trend of multibody modeling and provides a field for developing a technique to automate the composition of motion equations for mechanical systems. The proposed method should be further improved in the complex modeling of all-wheel-drive transmissions with several DMs.

**Author Contributions:** Conceptualization, M.D. and S.M.E.; methodology, M.D.; software, M.D.; validation, M.D. and S.M.E.; formal analysis, S.M.E.; investigation, M.D.; resources, S.M.E.; data curation, M.D.; writing—original draft preparation, M.D.; writing—review and editing, S.M.E.; visualization, M.D.; supervision, S.M.E.; project administration, S.M.E.; funding acquisition, S.M.E. All authors have read and agreed to the published version of the manuscript.

**Funding:** This research is financially supported by the Natural Sciences and Engineering Research Council of Canada (grant No. RGPIN-2020-04667).

**Institutional Review Board Statement:** No institutional review is required.

**Informed Consent Statement:** Not applicable.

**Data Availability Statement:** Some or all data, models, or code that support the findings of this study are available from the corresponding author upon reasonable request.

**Acknowledgments:** The authors are grateful to two anonymous reviewers for thorough and most helpful comments.

**Conflicts of Interest:** The authors declare no conflict of interest.

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
