# Peer review of "Improved Mathematical Approach for Modeling Sport Differential Mechanism"

_vehicles, doi:10.3390/vehicles4010005_

Round 1

Reviewer 1 Report

This paper proposed a mathematical modeling and simulating approach for the sport differential mechanism with controllable torque redistribution, which is confirmed efficiency by simulating in four conditions. The work carried out in this manuscript is meaningful and the methodology of the study is eligible. However, this manuscript is not yet ready for publication. Some specific comments on the content are listed below for future revision work. 1. The manuscript mentioned the developed mathematical tools effectively reduces the total number of equations for compactness and lowers the simulation time. But it’s not well exhibit in the manuscript, which makes the second conclusion not credible enough. 2. Table 1 listed numerous works just to prove that the research field of DM model is quite broad but there is lack of research on modeling of active differentials. I’m not sure if a table of published literature is enough or proper to obtain such a conclusion. 3. Figure 5 is not clear enough and there is no figure indicating kinetic parameters. 4. Some simulation results look weird, such as the switches of power direction in figure 7 and the violent shaking of M86 in figure 8. Why is that? These strange points need further explanation. 5. Typos: Section 2 is missing.

Author Response

Thank you so much for your comments. the paper was substantially revised to address your comments.

Reviewer 2 Report

A study on mathematical modeling to simulate the sport differential mechanism is presented in this article. The overall concept seems to be interesting. However, the writing suffers from various aspects from a scientific journal publication perspective.

The following comments and observations shall be considered.

  • The overall length of the article demands tedious efforts to perceive the work presented in this article. Consider trimming the paper by removing some of the contents with less relevance if applicable.
  • The abstract seems to be out of order. Reconsider reframing the statements used. In addition, the literature review mentioned in the abstract is nowhere in the rest of the article.
  • Section 2 seems to be missing. (Is it supposed to be the literature review?). Utmost care shall be taken to avoid these kinds of errors.
  • The introduction section does not possess statements to highlight the problem statement (though it is used later). It seems to be very vague.
  • How well is section 3, 4, and 5 justified to highlight the contribution of the proposed work? Again, it lacks clarity for using these sections, and it merely looks like a literature review.
  • The overall format seems not to match the journal template. In addition, all the references are out of order.
  • Avoid using abbreviations before they are expanded — for instance, ADS in the abstract.
  • Check all the figures and table labels carefully and their respective usages in relevant content.

Author Response

(The authors gave the same response as above.)

Round 2

Reviewer 2 Report

The authors have incorporated the suggested changes wherever applicable. The revised article looks clear and better compared to the first version. 

I would suggest the authors consider critical formatting corrections (including references) and use better resolution images.